# MobileWizard: A Data-Efficient GUI Agent with Structured Reasoning and Progressive Reinforcement Learning

## Abstract

This paper introduces **MobileWizard**, a data-efficient framework designed to enhance the reasoning and precision of mobile GUI agents. Trained on merely 24.5k public trajectories and 300 remedial trajectories, MobileWizard-7B demonstrates exceptional performance, achieving a 47.2% success rate on AndroidWorld, outperforming prominent larger open-source models like UI-TARS-72B. This high efficiency stems from two core innovations: 1) **Structured Reasoning**: A new structured Chain-of-Thought (CoT) paradigm that decomposes the agent's reasoning process into four explicit and interpretable modules: self-verification, screen analysis, planning, and action guidance. The proposed CoT guides the LLM to achieve logical consistency, extraction of key insights, and provides clear paths for failure analysis. 2) **Progressive Reinforcement Learning**: We propose a comprehensive RL strategy that features four key components: efficient cold-start training, a dynamic reward system with Progressive Reward Shrinking to boost precision, History Self-Alignment to narrow the training-inference gap, and a Corrective Teaching Pipeline for self-improvement from online failures. The experimental results demonstrate that our framework enables superior generalization from limited data. We believe that our method presents a scalable and efficient path toward building more robust and versatile GUI agents.

## 1 Introduction

The emergence of Vision-Language Models (VLMs) (Bai et al., 2025; Wang et al., 2024c; OpenAI, 2024; Google, 2024; Lu et al., 2024) has ignited a new wave of research into end-to-end GUI agents that can autonomously operate mobile applications from raw pixel inputs (Xu et al., 2024; Qin et al., 2025). These agents have achieved remarkable success by leveraging the powerful perception of VLMs to translate natural user commands into sequences of clicks, scrolls, and types. This approach holds immense promise for enhancing productivity and creating more intuitive user experiences across real-world mobile applications.

However, achieving a practical automated mobile agent still faces significant challenges. First, the data quality and availability pose a major hurdle. Existing open-source mobile GUI operation trajectory datasets are not only fewer than 50k samples—but also suffer from inconsistent quality. While manual annotation is an alternative, it is prohibitively expensive both in time and labor. Some efforts (Ye et al., 2025; Seed, 2025; Wang et al., 2025a) attempt to generate interaction data automatically by building large-scale environment infrastructures, but this approach also demands substantial hardware resources and reliable verification methods, limiting its accessibility for majority of researchers. Second, the reasoning paradigm of existing agents is not well optimized for GUI scenarios. Most agents' reasoning is plain and unconstrained, which leads to two critical flaws: (1) The thought process can become self-contradictory, failing to maintain a coherent analysis and plan across multi-step tasks. (2) When a task fails, the free-form reasoning text makes it hard to perform effective failure analysis. Third, current training methodologies used to build these agents are simple and suboptimal. Supervised Fine-Tuning (SFT) trains a model to replicate one specific ground-truth action, failing to recognize that multiple actions can be equally valid. While recent methods (Luo et al., 2025; Lu et al., 2025b; Zhou et al., 2025) have explored to adopt Reinforcement Leanring (RL), most of them focus solely on UI grounding, and some extend to task execution (Shi et al., 2025;

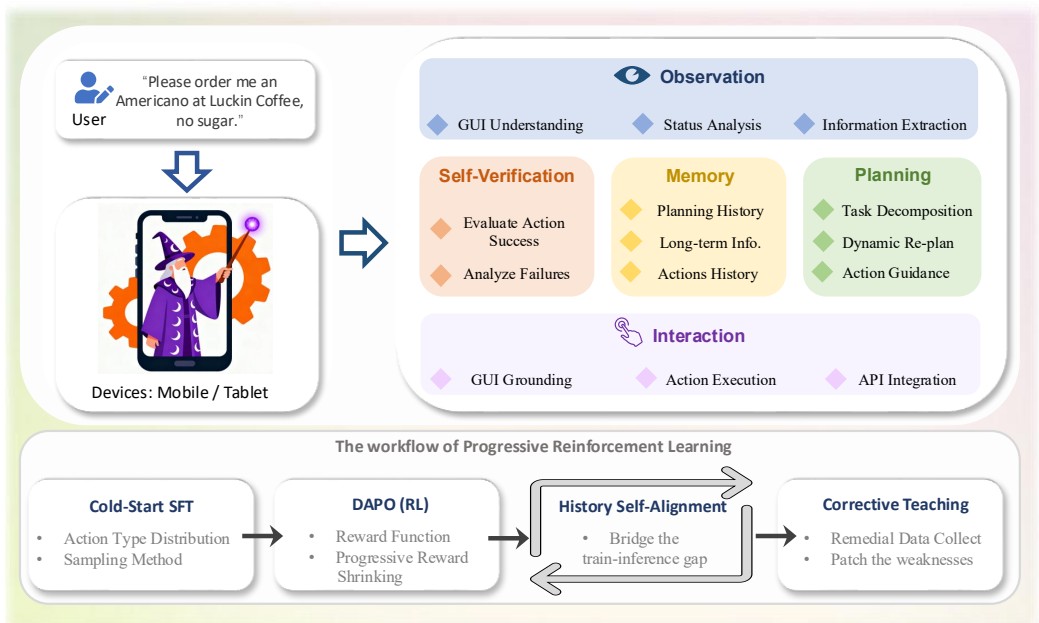

Figure 1: An Overview of the MobileWizard Framework.

Gu et al., 2025a; Seed, 2025) are constrained by simplistic reward functions and naive algorithmic adaptations, thus failing to harness the true capability of RL for intricate decision-making.

To address these challenges, we introduce MobileWizard, a mobile agent designed for high-precision GUI automation in a data-efficient manner. MobileWizard unifies perception, reflection, memory, planning, and grounding within a single end-to-end model. Our approach is centered around two innovations: *(1) Structured Reasoning.* To tackle the inherent reasoning deficiencies of general agents, we propose a structured Chain-of-Thought (CoT) paradigm. It compels the agent to decompose its reasoning into several explicit, interpretable modules: self-verification, screen analysis, planning, and action guidance. This structure enforces thinking on critical aspects, preventing the omissions or verbosity common in unstructured thought. It also facilitates the extraction of key insights, which are used to populate the agent's memory for subsequent steps, and improves logical coherence and provides transparent, debuggable reasoning pathways. *(2) A Progressive RL strategy.* It is composed of four integrated stages. First, it optimizes the initial cold-start training by determining the ideal data volume and action type distribution. Second, it employs a sophisticated reward function for both reasoning and actions, complemented by a Progressive Reward Shrinking mechanism that sharpens action precision by gradually tightening reward boundaries. Third, it narrows the gap between training and inference through History Self-Alignment, which encourages the model to take actions that are consistent with its past decisions, enhancing long-term task coherence. Finally, a Corrective Teaching pipeline enables continuous improvement by analyzing online failures to create small, high-impact remedial datasets, allowing the agent to efficiently learn from its mistakes and patch its weaknesses.

We evaluated MobileWizard across a wide spectrum of mobile GUI automation tasks, from GUI grounding to multi-step execution. Specifically, despite being trained on merely 24.5k public and 300 remedial trajectories, MobileWizard-7B achieves a 47.2% success rate on the challenging AndroidWorld benchmark and outperforms most leading agents on AndroidControl and GUI-Odyssey. Furthermore, it excels in GUI grounding, attaining 94.1% accuracy on ScreenSpot-v2 and 84.3% on MMBench-GUI, establishing it as a top-tier open-source model for this task.

## 2 RELATED WORKS

**GUI Agent Frameworks.** Research in GUI automation primarily follows two paradigms: multi-agent frameworks and end-to-end native agents. Multi-agent frameworks tackle complexity through decomposition. Systems like the Mobile-Agent series (Wang et al., 2024b;a; 2025c) employ a modular architecture with specialized agents for planning, execution, and reflection. Similarly, Agent-S (Hu

et al., 2024; Agashe et al., 2025) assigns distinct roles to agents to handle specific UI interaction patterns. This approach excels at intricate tasks by breaking them into manageable sub-problems. However, this modularity introduces significant system complexity and computational overhead from inter-agent coordination. In contrast, end-to-end agents represent a shift towards unified models that learn directly from raw pixel inputs. Pioneering models like CogAgent (Hong et al., 2024) leverage VLMs to integrate perception, reasoning, and action generation within a single network, offering architectural simplicity and high efficiency. Subsequent work (Gu et al., 2025b; Wang et al., 2025a; Ye et al., 2025) has further improved with techniques such as data cleaning and iteratively interaction data generation. However, the power of these models often stems from private trajectory data, which is both resource-intensive and demands manual effort. Therefore, MobileWizard explores an alternative direction, aiming to provide with deeper insights into model reasoning paradigm and training strategies to enhance the performance of GUI agents.

**Reasoning with GUI Agents.** Explicit reasoning is increasingly recognized as crucial for end-to-end GUI agents. Early works like CoAT (Zhang et al., 2024), Aguvis (Xu et al., 2024), and UI-TARS (Qin et al., 2025) initiated this by training models on data annotated with Chain-of-Thought (CoT). By learning to generate step-by-step thoughts before acting, these SFT-based methods induced basic planning abilities and achieved noticeable performance gains. However, this reasoning is fundamentally passive: the model merely mimics static thought patterns from annotations, rather than dynamically adapting its strategy based on real-time task outcomes. A major shift occurred with the adoption of RL, largely inspired by the "R1-style" paradigm (Guo et al., 2025). This approach allows agents to directly optimize their policies from task success signals, transitioning from "learning to imitate" to "learning to succeed." The benefits were substantial: GUI-R1 (Luo et al., 2025) achieved superior results with a fraction of the SFT data, while subsequent works introduced algorithmic refinements. For instance, Mobile-R1 (Gu et al., 2025a) and MobileGUI-RL (Shi et al., 2025) enhanced online interaction and optimized for both success and efficiency.

## 3 METHOD

This section details the MobileWizard framework. We begin by outlining our data statistics and preprocessing methods (Sec. 3.1). Next, we define a unified action space that equips the agent with both operational and question-answering capabilities (Sec. 3.2). Subsequently, we present our structured reasoning paradigm (Sec. 3.3) and Progressive Reinforcement Learning (Sec. 3.4).

### 3.1 DATA STATISTICS AND PREPROCESSING

As shown in Table 1, we collect five open-source datasets to ensure broad coverage of tasks and UI styles, including GUI-Odyssey (Lu et al., 2025a), AITZ (Zhang et al., 2024), AndroidControl (Li et al., 2024), AMEX (Chai et al., 2024), and GUI-Act (Chen et al., 2024). A primary limitation of these datasets is the general lack of explicit action descriptions, which are vital for an agent's reasoning and memory capabilities. To address this, we employed a powerful VLM to automatically generate a detailed action description for each step. Furthermore, recognizing that precise GUI grounding is critical for successful task execution, we repurposed our newly annotated data by extracting a subset of 'CLICK' actions. For each action, we create a training pair of $(Action\_Description, CLICK[x, y])$, which is used to enhance the model's ability to accurately map natural language descriptions to precise UI coordinates.

### 3.2 UNIFIED ACTION SPACE

A key challenge in existing datasets is that question-answering (QA) trajectories often lack explicit ground-truth answers, typically ending on a screen without a labeled answer text. To address this, we augment the action space defined by Qwen2.5-VL (Bai et al., 2025) with a new '*Answer*' action, and developed a robust pipeline to generate these missing labels: We first classifies tasks into manipulation or question-answering. For QA tasks, a powerful VLM analyzes the sequence of screenshots, extracts relevant information, and synthesizes a final answer. A human-in-the-loop process then verifies the VLM's output to ensure high accuracy of the annotations. This method allows us to generate explicit '*Answer (content)*' labels for these tasks, empowering the agent to move beyond operation and learn

Table 1: Statistics of the public datasets used for training, post-filtering for simplicity and redundancy.

| Type | Dataset | | | | | Overall |
| --- | --- | --- | --- | --- | --- | --- |
| | GUI-Odyssey | AITZ | AndroidControl | AMEX | GUI-Act | |
| # Steps | 70k | 3k | 71k | 24k | 15k | 183k |
| # Trajectories | 5.8k | 0.6k | 11k | 3k | 4.1k | 24.5k |

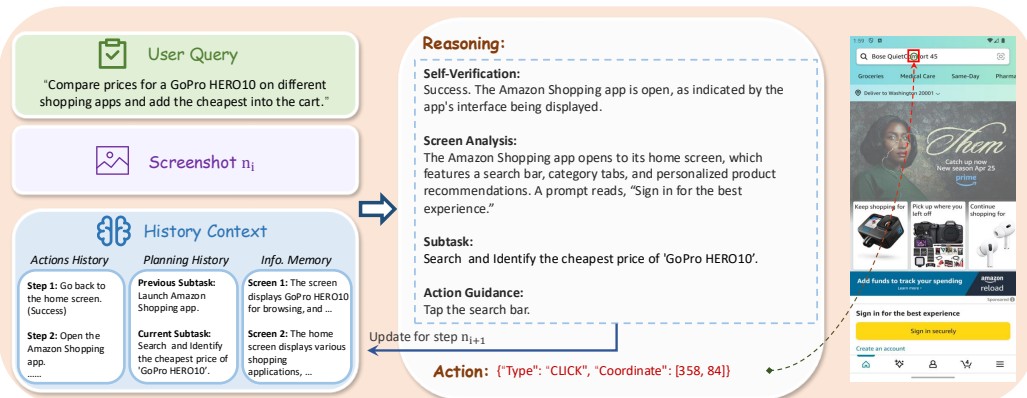

Figure 2: Illustration of the action-making workflow of MobileWizard.

to deliver conclusive, information-rich information. The complete set of actions and their definitions used in MobileWizard is detailed in Table 5.

### 3.3 STRUCTURED REASONING

Effective human-computer interaction on mobile devices follows a fundamental cognitive loop: perceiving the screen, planning the next step, executing an action, and verifying the result. To build a robust GUI agent, we explicitly model this process through our Structured Reasoning paradigm. This paradigm formalizes the agent's cognition into four distinct yet interconnected modules:

• **Screen Analysis** ($o_i$): For each step, the agent analyzes the screenshot to identify important UI elements and extract critical information relevant to the task instruction. This module forces the agent to firstly build a comprehensive understanding of the screen's state before making a decision.

• **Planning** ($p_i$): Based on the screen analysis and the task goal, the planning module formulates the immediate sub-task. Unlike a single, primitive action, a sub-task represents a higher-level objective that defines the direction of progress. This sub-task decomposition prevents the model from getting lost in long-horizon tasks.

• **Action Guidance** ($g_i$): This guidance serves as a direct and unambiguous command for the agent's current action, precisely specifying what to execute. It includes the action type and all necessary parameters to guide the agent in executing the command with high fidelity.

• **Self-Verification** ($v_i$): To imbue the agent with reflective capabilities, the self-verification module assesses the outcome of the previous action. It determines whether the action was successful, if the screen has changed as expected, and whether the current state is closer to achieving the sub-goal. This feedback loop allows the agent to recognize its own errors and adjust its plan accordingly.

We formulate a trajectory $\mathcal{T}$ as a sequence containing the initial user instruction followed by a series of steps. The full trajectory is defined as:

$$\mathcal{T} = (\text{instruction}, ([v_1, o_1, p_1, g_1], a_1), [v_2, o_2, p_2, g_2], a_2), \ldots, [v_n, o_n, p_n, g_n], a_n)) \quad (1)$$

where $a_n$ is the ground-truth action. Furthermore, historical context is critical for successful task execution. While many approaches use a simple history of past actions, we introduce a more structured memory system to provide a richer context. Specifically, when predicting the $n$-th step, the historical context $H_{n-1}$ from the previous $n-1$ steps is formulated as a tuple of three distinct

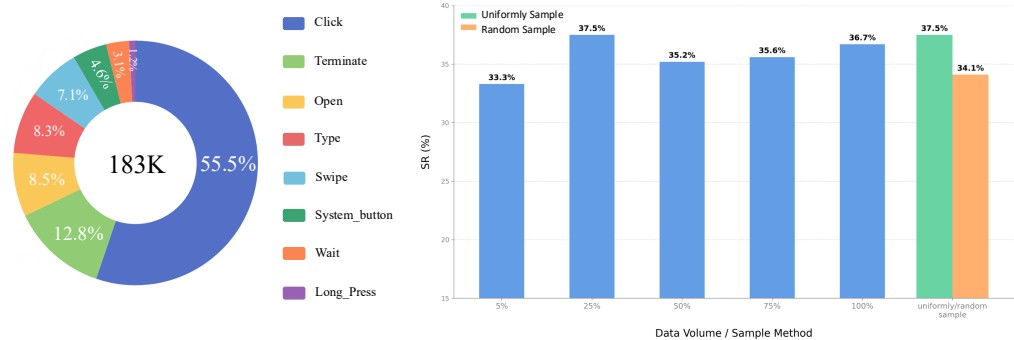

Figure 3: **(Left)** Action type distribution in our training dataset. **(Right)** Effect of data volume and sampling strategy on cold-start stage.

memory units:

$$H_{n-1} = (M_{n-1}^{\text{long\_info}}, M_{n-1}^{\text{action}}, M_{n-1}^{\text{plan}}) \tag{2}$$

The first unit, **Long-Term information Memory** ($M_{n-1}^{\text{long\_info}}$), acts as a persistent repository for critical visual information discovered throughout the entire trajectory. This includes critical state transitions and information highly relevant to the user query. This memory allows the agent to recall vital information even if it is no longer visible on the screen. The information stored here is primarily sourced from the *Screen Analysis*. The **Action History** ($M_{n-1}^{\text{action}}$), maintains an ordered sequence of all past *Action Guidance*s, formulated as $M_{n-1}^{\text{action}} = (g_1(s), g_2(s), \ldots, g_{n-1}(s))$, where $s$ represents the outcome status (success or failure) determined during *Self-Verification*. This provides the agent with a concrete record of what operations it has already performed. The third unit, **Planning History** ($M_{n-1}^{\text{plan}}$), stores the sequence of high-level sub-tasks, $M_{n-1}^{\text{plan}} = (p_1, p_2, \ldots, p_{n-1})$, which is crucial for maintaining strategic coherence and aligning current actions with established sub-goals. This comprehensive context $H_{n-1}$, together with the initial task instruction and the current UI screenshot, forms the complete input to the agent for the $n$-th step prediction. (We provide more details about the annotation process in Appendix C.1. We also analyze the key advantages of our structured paradigm compared to free-form reasoning in Appendix C.2.)

### 3.4 Progressive Reinforcement Learning

We propose Progressive Reinforcement Learning built upon the DAPO (Yu et al., 2025a) algorithm. It is composed of four synergistic components: the efficient data curriculum for cold-start, a sophisticated reward function with a Progressive Shrinking mechanism, the History Self-Alignment to bridge the train-inference gap, and a Corrective Teaching Pipeline.

### 3.4.1 Efficient Data Curriculum for Cold-Start

The initial cold-start phase is critical for establishing a strong baseline before RL. We identified two key factors for an effective cold-start: data volume and distribution. First, to determine effective data volume, we conducted an ablation study by performing SFT on varying subsets of the training data (5%, 25%, 50%, 75%, and 100%), each followed by 1 epoch of RL. As shown in the Figure 3, training under 25% data achieves the highest SR. We hypothesize that insufficient data fails to establish the fundamental reasoning paradigm, while excessive training leads to overfitting, which stifles the model's ability to explore diverse, high-reward trajectories during the RL phase. Second, to address the long-tail distribution of actions in open-source datasets, we employ uniform sampling over action types for the cold-start phase. A naive random sampling approach would leave low-frequency actions—such as *Long_Press*, *Answer*, and *System_Button*—severely undertrained, compromising the model's ability to execute them during subsequent RL. The results confirm that uniform sampling provides a superior performance (37.5%) compared to the random sampling baseline (34.1%).

### 3.4.2 Reward Function

In the RL setup, our reward function has four main parts:

**Format Reward.** The format reward ($R_{format}$) validates two structural properties of the output. First, it ensures the output adheres to the specified tag order: a reasoning block enclosed by <think> tags, followed by an action block within <answer> tags. Second, it verifies that the reasoning block contains our four proposed parts, arranged in the correct sequence.

**Action Type Reward.** The Action-type reward $R_{type}$ is 1.0 if the predicted type matches the ground truth, and 0.0 otherwise. If the types differ, all associated parameter rewards are also set to zero.

**Parameters Reward.** We divide action parameters reward, $R_{param}$, into two groups: (1) Grounding parameters for actions like *Click* and *Long_Press*, which must output screen coordinates. (2) Text parameters for actions like *Type*, *Open_App* and *Answer*, which must output a content.

For grounding, we base the reward on the distance between the predicted and ground-truth coordinates. However, we found that picking a fixed distance threshold is tricky. If the threshold is too small, early rewards are almost always zero, giving little learning signal; In contrast, if the threshold is too large, the model can engage in reward hacking by clicking outside the target element yet still earning reward. To resolve this dilemma, we propose **Progressive Shrinking** mechanism. As training proceeds, the allowed threshold gradually tightens—for example, shrinking from 14% of the screen resolution, to 10%, then 7%, and finally 4%. This forces the model to incrementally refine its pointing precision, evolving from coarse localization to high-fidelity grounding. For textual actions, the reward is derived from the token-level F1-score. To enforce a baseline quality, this reward is gated: it is set to zero for predictions with an F1-score below 0.5, and equals the F1-score itself otherwise.

**Reasoning Reward.** Free-form reasoning presents a significant challenge for reward modeling. It is often verbose and highly variable, making it intractable to validate the quality of the reasoning process. In contrast, our structured reasoning paradigm, which comprises four explicit and goal-oriented modules, offers a tractable solution. The concise and well-defined nature of each module's content allows us to assess the quality of each reasoning step independently. To do this, we compute a semantic similarity score between the generated content of each part and its corresponding ground-truth annotation. We use the Sentence-BERT score (Reimers & Gurevych, 2019) as our similarity metric, implemented with the Qwen3-Embedding-8B model (Zhang et al., 2025). The final reasoning reward, $R_{reasoning}$, is the average of four module-specific scores.

Intergrating all reward components, the final reward $R_{\text{total}}$ is computed as:

$$R_{\text{total}} = w_1 \cdot R_{\text{format}} + w_2 \cdot R_{\text{reasoning}} + w_3 \cdot (R_{\text{type}} + R_{\text{param}}) \tag{3}$$

where $w_1$, $w_2$ and $w_3$ are hyperparameters that balance the relative importance of quality and execution correctness.

### 3.4.3 History Self-Alignment

A significant challenge in training multi-step GUI agents is the discrepancy between the supervised training setup and real-world inference. During training, the model learns to predict the next action based on a ground-truth annotated historical context. During inference, however, it must rely on its own previously generated content where any deviation can propagate and lead to task failure. This phenomenon, known as exposure bias, can severely limit the model's robustness. To mitigate this, recent works (Tao et al., 2024; Gu et al., 2025b) have explored aligning training histories with the model's own reasoning patterns. Building on this direction, we introduce a more strategic and computationally efficient History Self-Alignment mechanism that is deeply integrated with the RL process. Our alignment mechanism is not executed at a fixed frequency but is instead bottleneck-driven, activating only when the model's validation reward stagnates

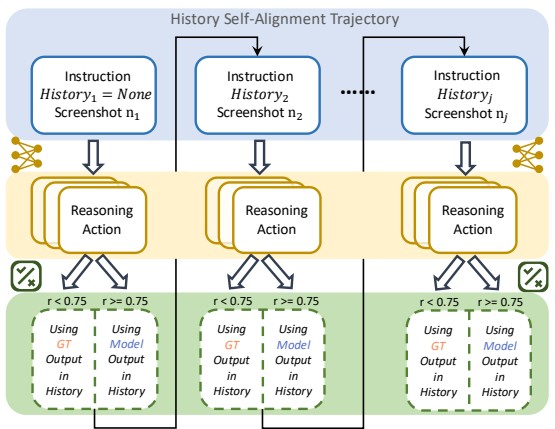

Figure 4: The **History Self-Alignment** process refines the historical context step-by-step, creating a new trajectory with updated history for the next training.

for $N$ steps. This strategy minimizes computational overhead by avoiding alignment when the model is learning effectively. Upon triggering, the process operates on a randomly sampled 50% subset of the training data to further manage costs. For the core alignment procedure, we first prompt the current policy model to generate $N$ diverse candidate output for each training sample by using varied temperatures. As shown in Figure 4, our method operates at the trajectory level: each output is generated autoregressively, conditioned on the model's own preceding history or the ground-truth, thereby updating the whole history. This contrasts with step-level approaches like UI-Venus, which only modify the latest step content in history. Based on our proposed reward function, model's output is accepted only if its reward exceeds a predefined threshold (0.75). If all $N$ candidates fail to meet this threshold, the original labeled content is retained. Furthermore, if multiple candidates surpass the threshold, we filter for diversity by using Sentence-BERT scores to remove content that are too semantically similar to each other. By iteratively applying this process, we force the model to continuously learn from its own increasingly competent and diverse reasoning. This transforms the static dataset into a "living" curriculum that dynamically adapts to the agent's evolving capabilities.

### 3.4.4 CORRECTIVE TEACHING PIPELINE

To further address model failures observed during online inference, we introduce the Corrective Teaching Pipeline, a semi-automated system designed to diagnose and patch agent weaknesses. This pipeline generates targeted remedial data from the agent's online rollouts:

- **Failure Diagnosis**: Each online rollout is first evaluated by Gemini-2.5-Pro to classify it as a Success or Failure. Successful trajectories are directly added to the training pool. For failed trajectories, the verifier is prompted to identify the erroneous step(s) that led to the failure.

- **Teacher-Guided Repair**: Beginning from the identified point of failure, we invoke a capable teacher model (UI-Tars-1.5 (Seed, 2025)) to generate a corrective action sequence and complete the remainder of the task. This process grafts the teacher's expert actions onto the agent's correct initial steps, forming a new hybrid trajectory. This repaired trajectory is then re-submitted to the verifier. If validated as successful, it is accepted as a corrective example; otherwise, it is flagged for manual annotation by a human expert, ensuring no failure is left unaddressed.

By iteratively feeding this high-quality, diversified remedial data back into the RL loop, our agent efficiently learns from its specific shortcomings. Promisingly, our experiments demonstrate that as few as 300 remedial trajectories are sufficient to drive significant and targeted improvements.

## 4 EXPERIMENTS

### 4.1 IMPLEMENTATION DETAILS

To thoroughly assess MobileWizard's performance, it is evaluated across a wide range of mobile GUI benchmarks. Our MobileWizard-7B model is initialized from the Qwen2.5-VL-7B (Bai et al., 2025). We utilize the Android Studio emulator as the interactive environment for our Corrective Teaching Pipeline. More training details can be found in Appendix B.

### 4.2 EXPERIMENTAL RESULTS

### 4.2.1 GUI GROUNDING BENCHMARKS

We evaluated MobileWizard's grounding capabilities on two widely-recognized benchmarks: ScreenSpot-v2 (Wu et al., 2024) and MMBench-GUI-L2 (Wang et al., 2025b). ScreenSpot-v2 is a general-purpose benchmark that tests text and icon localization across mobile, desktop, and web platforms. MMBench-GUI-L2 assesses real-world performance using a broad range of diverse and challenging queries. For all evaluations, we followed the standard protocol: a prediction is considered correct if its coordinate falls within the ground-truth bounding box of the target element.

As shown in Table 2, MobileWizard-7B demonstrates exceptional performance. On ScreenSpot-v2, our model achieves SOTA overall score of 93.0, as the same as UI-TARS-7B. On the more challenging MMBench-GUI-L2 benchmark, MobileWizard-7B continues to showcase its strong capabilities, attaining an overall score of 79.8. Our model substantially outperforms most of similarly-sized

Table 2: Comparison on ScreenSpot-v2 and MMBench-GUI-L2 grounding benchmarks. Underlined denotes the second-best open-source performance.

| Model | ScreenSpot-v2 | | | MMBench-GUI | | | | |
| | Mobile | | Overall | iOS | | Android | | Overall |
| | Text | Icon | | Basic | Adv. | Basic | Adv. | |
| --- | --- | --- | --- | --- | --- | --- | --- | --- |
| *Close-Source Models* | | | | | | | | |
| GPT-4o (OpenAI, 2024) | 26.6 | 24.2 | 25.4 | 5.1 | 3.3 | 2.5 | 1.4 | 3.1 |
| Qwen-Max-VL (Bai et al., 2025) | - | - | - | 77.3 | 59.0 | 79.5 | 70.1 | 71.5 |
| *Open-Source Models* | | | | | | | | |
| Qwen2.5-VL-7B (Bai et al., 2025) | 97.6 | 87.2 | 92.4 | 66.5 | 55.1 | 35.1 | 35.2 | 48.0 |
| Aguvis-7B (Xu et al., 2024) | - | - | - | 67.5 | 65.1 | 60.9 | 50.9 | 61.1 |
| UGround-7B (Gou et al., 2024) | 75.1 | 84.5 | 79.8 | **92.7** | 70.9 | **93.5** | 71.0 | 82.0 |
| OmniParser-v2 (Yu et al., 2025b) | 95.5 | 74.6 | 85.1 | - | - | - | - | - |
| OS-Atlas-7B (Wu et al., 2024) | 95.2 | 75.8 | 85.5 | 74.8 | 48.8 | 69.6 | 46.7 | 60.0 |
| UI-TARS-7B (Qin et al., 2025) | 96.9 | 89.1 | **93.0** | - | - | - | - | - |
| UI-TARS-72B (Qin et al., 2025) | 94.8 | 86.3 | 90.6 | 90.7 | **81.2** | 92.9 | **80.0** | **86.2** |
| UI-TARS-1.5-7B (Seed, 2025) | - | - | - | 88.5 | 69.3 | 90.4 | 69.3 | 79.4 |
| MobileWizard-7B (ours) | 98.1 | 87.9 | **93.0** | 89.3 | 72.2 | 89.1 | 72.6 | 79.8 |

models, including OS-Atlas-7B (60.0) and UI-TARS-1.5-7B (79.4). This performance underscores the effectiveness of our training methodology, indicating a inspiring balance between high-performance grounding and data efficiency.

### 4.2.2 TASK EXECUTION BENCHMARKS

To evaluate MobileWizard's task execution capabilities, we assess its performance on online and offline evaluation scenarios to provide a holistic view of the agent's abilities.

**Offline Evaluation.** On the AndroidControl-High benchmark (Li et al., 2024), MobileWizard-7B achieves a step Success Rate (SR) of 74.9%, outperforming most mainstream agents and performing only marginally lower than the leading model, UI-Venus-Navi-7B. On AndroidControl-Low, our model sets a new SOTA with a 94.6% step SR, surpassing the previous top model, UI-Genie-7B (94.3%), which was trained on more data. Furthermore, on GUI-Odyssey (Lu et al., 2025a), MobileWizard-7B demonstrates strong performance with 92.8% type accuracy and 83.3% step SR, confirming its robustness across diverse offline tasks.

**Online Evaluation.** We evaluate real-world task execution on AndroidWorld (Rawles et al., 2025), an online benchmark requiring continuous interaction with live mobile applications. As shown in Table 3, MobileWizard-7B achieves a 47.2% Success Rate (SR). This result not only establishes a new SOTA for models trained on public data but also surpasses prominent models trained on extensive proprietary datasets, including UI-Genie-7B (36.3%) and UI-TARS-72B (46.6%). Notably, our model's performance is only 1.9% lower than the leading model, demonstrating the exceptional data efficiency and generalization of our approach.

### 4.3 ABLATION STUDY

• **Impact of the Reasoning Paradigm.** We first investigate our structured reasoning paradigm. We evaluate two more variants: Free-form Reasoing and No Reasoning baseline. As shown in Table 4(a), the Free-form Reasoning model offers only a marginal improvement over the No Reasoning baseline, with both variants lagging significantly behind our proposed method. This demonstrates that while explicit reasoning is beneficial, its effectiveness is critically dependent on a structured format that enforces a systematic analysis, compelling the model to make more well-grounded decisions.

• **Effect of Reasoning Reward.** To isolate the effect of the reasoning reward, we trained two agents from the same initial policy: one with the reward and one without. The inclusion of the reasoning reward significantly improved the performance, confirming that explicitly rewarding high-quality reasoning is an effective strategy for refining the agent's internal thought process.

Table 3: Performance comparison on GUI task execution benchmarks. 'AC' means AndroidControl.

| Models | Offline | | | | | | Online |
|---|---|---|---|---|---|---|---|
| | AC-Low | | AC-High | | GUI-Ody. | | AndroidWorld |
| | Type Acc. | SR | Type Acc. | SR | Type Acc. | SR | |
| **Closed-Source Models** | | | | | | | |
| GPT-4o (OpenAI, 2024) | 74.3 | 19.4 | 66.3 | 20.8 | 34.3 | 3.3 | - |
| SeedVL-1.5 (OpenAI, 2024) | - | - | - | - | - | - | 62.1 |
| UI-TARS-1.5 (OpenAI, 2024) | - | - | - | - | - | - | 64.2 |
| **Open-Source Models** | | | | | | | |
| *More Private Datasets* | | | | | | | |
| UI-TARS-7B (Qin et al., 2025) | 98.0 | 90.8 | 83.7 | 72.5 | 94.6 | 87.0 | 33.3 |
| UI-TARS-72B (Qin et al., 2025) | **98.1** | 91.3 | 85.2 | 74.7 | **95.4** | **88.6** | 46.6 |
| UI-Genie-7B (Xiao et al., 2025) | **98.1** | **94.3** | 83.5 | 74.2 | - | - | 36.3 |
| UI-Venus-Navi-7B (Gu et al., 2025b) | 97.1 | 92.4 | **86.5** | **76.1** | 87.3 | 71.5 | **49.1** |
| *Only Public Datasets* | | | | | | | |
| Qwen2.5-VL-7B (Bai et al., 2025) | 94.1 | 85.0 | 75.1 | 62.9 | 59.5 | 46.3 | 21.8 |
| SeeClick (Cheng et al., 2024) | 93.0 | 75.0 | 82.9 | 59.1 | 71.0 | 53.9 | - |
| GUI-R1-7B (Luo et al., 2025) | 85.2 | 66.5 | 71.6 | 51.7 | 65.5 | 38.8 | - |
| OS-Atlas-7B (Wu et al., 2024) | 93.6 | 85.2 | 85.2 | 71.2 | 84.5 | 62.0 | - |
| Aguvis-72B (Xu et al., 2024) | - | 84.4 | - | 66.4 | - | - | 26.1 |
| MobileGUI-7B (Tang et al., 2025) | - | - | - | - | - | - | 30.0 |
| MobileWizard-7B (ours) | 97.5 | **94.6** | 85.3 | 74.9 | 92.8 | 83.3 | 47.2 |

Table 4: Ablation studies. We report the Success Rate (SR, %) in AndroidWorld.

| Method | SR (%) |
|---|---|
| No CoT | 34.8 |
| Free-form | 35.4 |
| **Structured** | **37.5** |

(a) Reasoning Paradigm

| Method | SR (%) |
|---|---|
| w/o R.Reward | 36.3 |
| **w/ R.Reward** | **37.5** |

(b) Reasoning Reward

| Method | SR (%) |
|---|---|
| w/o HSA | 38.0 |
| **w/ HSA** | **40.8** |

(c) History Self-Alignment

| Method | SR (%) |
|---|---|
| w/o remedial | 45.6 |
| **w/ remedial** | **47.2** |

(d) Remedial Data

• **Effect of History Self-Alignment.** As shown in Table 4(c), incorporating history self-alignment significantly improves performance. This is achieved by progressively bridging the gap between training and inference. The mechanism encourages the model to take actions that are consistent with its past decisions, thereby enhancing long-term task coherence.

• **Effect of Remedial Data.** To evaluate the benefit of corrective feedback, we compared two settings: one with the training data supplemented by 300 remedial trajectories, and one without. The inclusion of this remedial data yielded a further significant performance boost (+1.6%), demonstrating the model's ability to correct its own flaws by learning from a minimal amount of targeted feedback.

## 5 CONCLUSION AND FUTURE WORK

In this paper, we introduced MobileWizard, a data-efficient framework designed to address key challenges in mobile GUI agent development. Our approach combines two core innovations: a Structured Reasoning paradigm that enhances the agent's interpretability and logical consistency, and a Progressive Reinforcement Learning strategy that effectively boosts action precision and enables self-correction. Experimental results validate that MobileWizard achieves outstanding performance on multiple benchmarks, despite being trained on a small, publicly available dataset. This work demonstrates that a well-designed framework can effectively overcome data limitations, presenting a scalable and efficient path toward building more robust GUI agents.

Looking ahead, we aim to explore two primary directions. First, we will investigate the impact of expanding the remedial dataset to further enhance the model's robustness. Second, to enhance the agent's ability to generalize to unseen mobile applications, we will investigate methods for autonomous exploration and real-time reward, allowing it to adapt and learn on its own.

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

# APPENDIX

## A    UNIFIED ACTION SPACE

Table 5: The Unified Action Space for MobileWizard. It covers standard UI interactions, system-level commands, and task-completion actions.

| Action | Definition |
|---|---|
| `OPEN_APP(app="")` | Launch the specified app. |
| `CLICK(x, y)` | Click at coordinates (x, y). |
| `LONG_PRESS(x, y)` | Long press at coordinates (x, y). |
| `SWIPE((x1, y1), (x2, y2))` | Swipe from (x1, y1) to (x2, y2). |
| `TYPE(content="")` | Type the specified content. |
| `WAIT(second="")` | Wait for loading. |
| `System_Button(Back)` | Press the system 'back' button. |
| `System_Button(Home)` | Press the system 'home' button. |
| `System_Button(Enter)` | Press the 'enter' key button. |
| `System_Button(Recent)` | Press the 'recent' button. |
| `Terminate(status="")` | Terminates the task. Status can be 'success' (task completed) or 'failure' (task failed). |
| `Answer(content="")` | Concludes an information-seeking task by providing the final answer. |

## B    TRAINING HYPERPARAMETERS

**Cold-Start.** We train the init agent based on Qwen2.5-VL-7B using the AdamW optimizer with a learning rate of $1 \times 10^{-5}$ and a global batch size of 128.

**RL.** Based on DAPO (Yu et al., 2025a) algorithm, this training stage is conducted with a learning rate of $1 \times 10^{-6}$ and a rollout sample size of 8. The global batch size is set to 192 for MobileWizard-7B. During the History Self-Alignment stage, the model is trained iteratively for 5 epochs. Finally, we incorporated remedial data and conducted one more training epoch to obtain the final model.

## C    MORE DETAILS OF STRUCTURED REASONING

### C.1    ANNOTATIONS METHOD

To construct our training dataset, we employ Gemini-2.5-Pro (Comanici et al., 2025), one of the most powerful VLM, to generate step-by-step structured reasoning. Our annotation methodology is trajectory-level, designed to mirror the model's sequential training and inference flow. Specifically, the reasoning generated for a given step is fed as historical context into the annotation process for the subsequent step.

At each step, the VLM is prompted with a rich set of inputs: three consecutive screenshots (previous, current, and next). The current screenshot is uniquely marked with a visual prompt (Yang et al., 2023; Lin et al., 2024) to highlight the exact location of the action. Leveraging this multi-frame context, the model generates four distinct components of reasoning: *(1) Self-Verification:* By comparing the previous and current screenshots, the model assesses the outcome of the prior action to confirm its successful execution. *(2) Screen Analysis:* The model analyzes the current screenshot to identify crucial UI elements and extract key information. *(3) Planning:* Drawing upon the three screenshots, its understanding of the workflow, and general world knowledge, the model decomposes the high-level task into sub-tasks and determines the most immediate sub-task to address. *(4) Action Guidance:* This is directly populated with the corresponding action description. To ensure data quality, the model also outputs a confidence score with each annotation. A score below a predefined threshold signals potential model confusion and unreliable reasoning. These instances are automatically flagged for

manual review and correction by a human annotator, implementing a human-in-the-loop verification process.

## C.2 STRUCTURED VS. FREE-FORM

Compared to free-form reasoning, our structured reasoning model presents three primary advantages:

• **Enhanced Information Traceability and Memory Consolidation:** Sequential action-making in GUI environments is contingent upon historical context. Our structured format facilitates seamless information extraction, allowing for the programmatic retrieval of specific analytical outcomes from prior steps. This information can then be systematically consolidated into a coherent memory for future decisions. Conversely, extracting salient information from plain, free-form text is a non-trivial parsing challenge.

• **Enforced Analytical Rigor and Consistency:** Our structured Chain-of-Thought (CoT) constrains the model's reasoning process, ensuring it consistently thinks in a set of pre-defined, critical dimensions. Free-form CoT, being unconstrained, suffers from high output variance; its reasoning can be overly laconic, thus omitting key insights, or excessively verbose, diluting important information with superfluous details.

• **Clear Debugging and Failure Attribution:** A key limitation of unstructured reasoning is its opacity during failure analysis. It is difficult to diagnose whether a failure originates from perception, planning, or execution. Our modular, structured framework provides inherent interpretability and a clear path for failure attribution. We can precisely diagnose the point of failure by examining each component's output. For instance, we can determine if an error stems from a perceptual mistake (e.g., Screen Analysis), a flawed strategy (e.g., Planning), or a malformed command (e.g., Action). This modularity is invaluable for targeted debugging and the iterative refinement of the model.

## D ILLUSTRATION OF TRAINING DYNAMICS

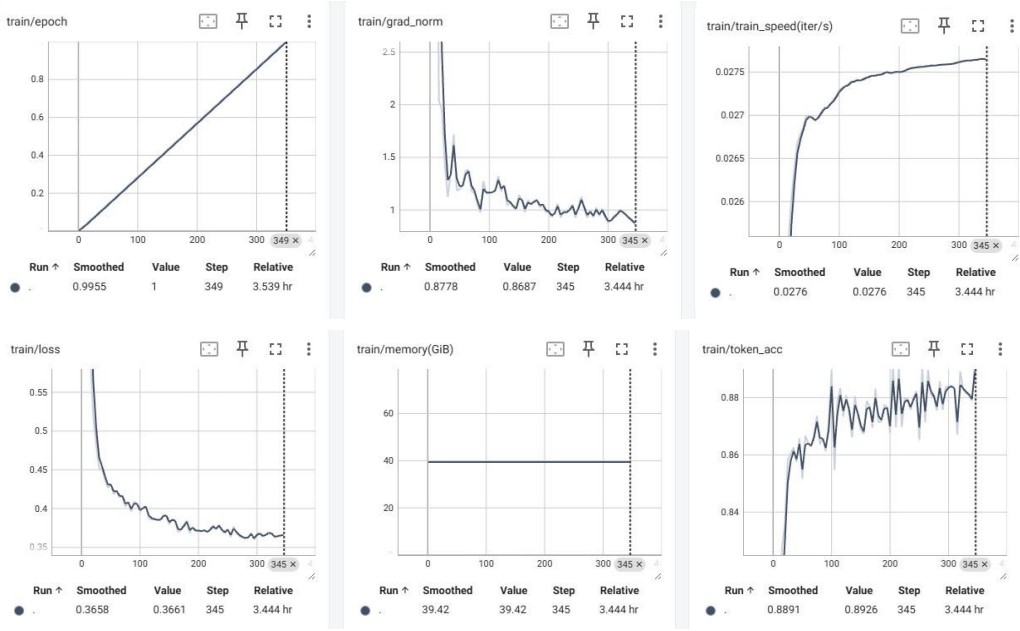

Figure 5: Training Dynamics in Cold-Start Stage, including loss, training speed, memory usage, token accuracy, and grad norm.

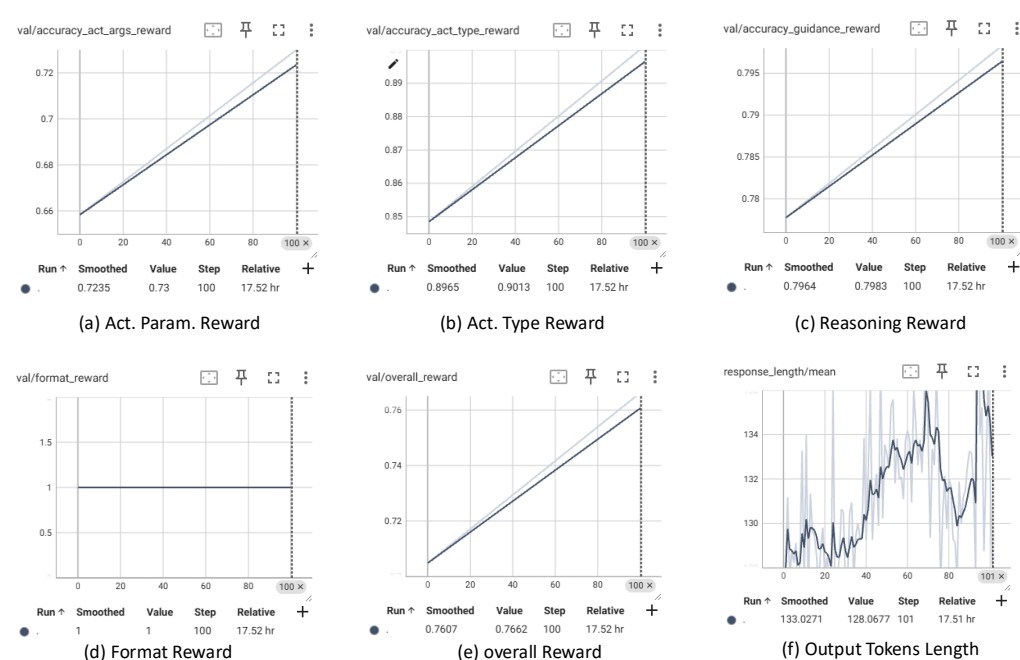

Figure 6: Training Dynamics in RL Stage on the **Validation Set**, including action parameters reward, action type reward, reasoning reward, format reward, and output tokens length.

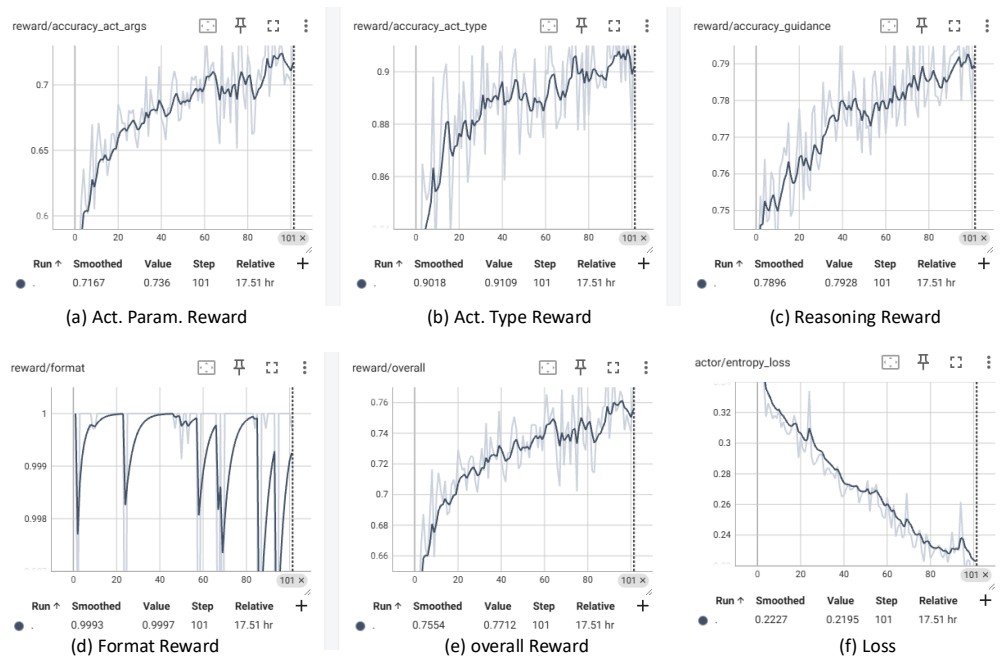

Figure 7: Training Dynamics in RL Stage on the **Train Set**, including action parameters reward, action type reward, reasoning reward, format reward, and loss.

