# OpenReview forum: "MobileWizard: A Data-Efficient GUI Agent with Structured Reasoning and Progressive Reinforcement Learning"
_ICLR.cc/2026/Conference — Submitted to ICLR 2026_

### Official Review · Reviewer_mgKA · 2025-10-20

**Soundness:** 1
**Presentation:** 2
**Contribution:** 1
**Rating:** 2
**Confidence:** 3

**Summary:**

This paper proposes a GUI agent framework named *MobileWizard*, which introduces two key innovations:
1. A structured Chain-of-Thought paradigm comprising four components:
   a. self-verification,
   b. screen analysis,
   c. planning, and
   d. action guidance.
2. A progressive reinforcement learning approach consisting of four components:
   cold-start training, a reward system with progressive reward shrinking, history self-alignment, and a corrective teaching pipeline.

**Strengths:**

The presentation is mostly clear. The two innovations are described step by step and in detail, and the evaluation effectively demonstrates the proposed approach’s performance, with both comparish with baselines and ablation study.

**Weaknesses:**

The motivation is unclear—it’s not evident what drives these two innovations, nor how they specifically benefit GUI agents in mobile environments.
Descriptions and explanations for Figures 1 and 2 are missing.
Some word choices are vague or unsupported. For example, phrases like “...for critical visual information...” and “This includes critical state transitions and...” (line 234) raise questions. What qualifies as “critical”? Are there non-critical transitions or information?

The claim that *MobileWizard-7B demonstrated exceptional performance* (line 375) is unconvincing for several reasons:
1. UI-TARS-7B achieves the same overall best performance on the ScreenSpot-v2 dataset.
2. UGround-7B outperforms MobileWizard-7B on the MMBench-GUI dataset (82.0% vs. 79.8%).
3. In Table 3, multiple baselines—including UI-TARS-7B and UI-Venus-Navi-7B—show better performance. Specifically, UI-Venus-Navi-7B achieves a 49.1% SR, compared to 47.2% from the proposed approach.
Given these results, how does the proposed approach qualify as “exceptional”?

**Typo:**
“We first classifies...” in line 158 should be corrected to “We first classify...”

**Questions:**

- How are subtasks derived from tasks? Is there a one-to-many relationship between tasks and subtasks? Are subtasks independent of each other?
- In lines 233–245, if Action History and Planning History both trace back to time index 1, aren’t they also long-term information? If so, the naming is unintuitive—“Long-Term Information Memory” implies the others are not long-term.

---

> ### Author Response · Authors · 2025-11-21
> **Response to Reviewer mgKA**
>
> ### **Response to W1: The motivation is unclear—it’s not evident what drives these two innovations, nor how they specifically benefit GUI agents in mobile environments.**
> Here, we will state again the motivation behind our two key innovations:
> 1. Structured CoT is designed to simulate a fundamental human cognitive loop, guiding the model’s reasoning process to deliberately incorporate certain critical aspects. This process mirrors a multi-agent framework, where specialized agents collaborate—a workflow we internalize within the end-to-end model's reasoning process. More importantly, structured CoT allows us to efficiently extract and organize key information in Memory from the reasoning content, thereby strengthening the model’s ability to retain and utilize historical context.
> 2. Progressive RL represents an enhancement to previous RL-based training strategies for GUI agents. We identified several seemingly simple yet important issues within the training process and proposed targeted solutions. These adjustments have led to a noticeable improvement in model performance.
>
> ### **Response to W2: Descriptions and explanations for Figures 1 and 2 are missing.**
> We apologize if the lack of detailed explanations in Figures caused any difficulty in understanding. We will add more explanatory notes in the revised version.
>
> ### **Response to W3: Some word choices are vague or unsupported. Phrases like “...for critical visual information...” What qualifies as “critical”? Are there non-critical transitions or information?**
> Here, we provide further explanation:
> During screen analysis, the screen state often contains a large amount of redundant visual information that is not directly relevant or helpful for accomplishing the goal instruction. Therefore, the model only needs a general understanding of the overall layout and this non-critical visual content. Conversely, it must carefully and thoroughly extract information that is directly relevant to the goal instruction and record it in the history memory. This includes important UI element states or key textual descriptions from images, which we refer to as critical visual information.
>
> ### **Response to W4: The claim that MobileWizard-7B demonstrated exceptional performance is unconvincing**
> We apologize for confusion from our wording. Our goal wasn’t to claim exceptional performance, but to show our two methods work well with limited training data—we used far less than typical GUI agents yet still got comparable, promising results. Scaling with more high-quality data later could boost performance (our next focus). We’ll revise misleading statements in the revision.
>
> ### **Response to Q1: How are subtasks derived from tasks? Is there a one-to-many relationship between tasks and subtasks? Are subtasks independent of each other?**
> First, let's clarify the difference between subtasks and the goal task. For example, if the goal task is:
> "Book a flight for two people from Beijing to Los Angeles, departing at 8:00 on November 20." Then the possible decomposed subtasks could be:
> 1. Open the flight booking app and navigate to the ticket purchase page.
> 2. Set Beijing as the departure city and Los Angeles as the destination.
> 3. Select the date and time: November 20, 8:00.
> 4. Search for and choose the most suitable flight.
> 5. Complete the booking process.
>
> As shown, the goal task can only be accomplished by finishing these subtasks step by step.
>
> - How are subtasks derived from tasks? During data construction, we leverage the strong general world knowledge of Gemini-2.5 Pro, prompting it to generate necessary subtasks for each goal task. It is worth noting that the process is not simply splitting the goal task text into several subtask texts. Instead, the model is provided with the ground-truth trajectory (each step’s state screenshot + executed action). This ensures that each generated subtask is more accurate and reasonable.
> - A goal task must correspond to multiple subtasks.
> - Each subtask is independent, with its own distinct objective. Only when all subtasks are completed is the goal task considered accomplished.
>
> ### **Response to Q2: if Action History and Planning History both trace back to time index 1, aren’t they also long-term information? If so, the naming is unintuitive—“Long-Term Information Memory” implies the others are not long-term.**
> Thanks for pointing this out. In fact, if a task requires multiple steps to complete, both Action History and Planning History can be classified as long-term history, since they are recorded starting from the first step. The difference between them and Information History is that action and plan records are one sentence per step, while information records are a paragraph. In the revised version, we will polish the relevant terms to ensure readers can understand more clearly.

---

> > ### Comment · Reviewer_mgKA · 2025-11-21
> > **Initial Response to Author Response**
> >
> > Thank you for your response, which clarifies many question I had. I have a few additional questions.
> >
> > On W1,
> >
> > 1. Can you point out where you have stated these two points in the manuscript?
> > 2. These two presented points in your response are the two innovations you proposed and their benefits. I was original asking about what led to them? For example, were they brought up because of inherent limitations in existing methods, that you have to have these two innovations? Or are there other reasons? Or did you brainstorm and come up with these two innovations and then find out they work? I was asking about what DRIVES these two innovations, not what your innovations are about (which are clear to me).
> >
> > On W3,
> >
> > Unless 'critical' will be interpreted in the same way as your definition for sure by the majority of readers, Ann explicit definition should be included in the manuscript. Namely - *This includes important UI element states or key textual descriptions from images, which we refer to as critical visual information.*
> >
> > On Q1,
> >
> > Can you point out the description of setup 'we leverage the strong general world knowledge of Gemini-2.5 Pro...' is stated in the manuscript?

---

> > > ### Author Response · Authors · 2025-11-21
> > > **Replying to Reviewer mgKA (1/2)**
> > >
> > > > ### **Response to Q1: Point out where you have stated these two points?**
> > >
> > > **Regarding Structured CoT:**
> > >
> > > - **On "simulating a fundamental human cognitive loop":**
> > >   This concept is introduced at the beginning of **Sec. 3.3 (Structured Reasoning)**: *"Effective human-computer interaction on mobile devices follows a fundamental cognitive loop: ......"*
> > >
> > > - **On "Multi-agent framework and Internalization":**
> > >   We first compared multi-agent frameworks with end-to-end models in **Related Works** and mentioned unifying these capabilities in the **Introduction**:
> > >   - **Line 106:** *"Multi-agent frameworks tackle complexity through decomposition... employ a modular architecture with specialized agents for planning, execution, and reflection."*
> > >   - **Line 80:** *"Mobile Wizard unifies perception, reflection, memory, planning, and grounding within a single end-to-end model."*
> > >   - Our structured reasoning decomposes thought processes into *"explicit, interpretable modules"*, which essentially internalizes the specialized division of labor found in multi-agent collaborations.
> > >
> > > - **On "Extract and organize key information in Memory":**
> > >   This is mentioned in the **Introduction** and detailed in **Appendix C.2**:
> > >   - **Line 85:** *"It also facilitates the extraction of key insights, which are used to populate the agent's memory for subsequent steps..."*
> > >   - **Line 709:** *"Our structured format facilitates seamless information extraction... systematically consolidated into a coherent memory for future decisions."*
> > >
> > > **Regarding Progressive RL:**
> > >
> > > - **On "Enhancement to previous RL-based training strategies":**
> > >   This viewpoint is primarily established in the **Introduction**.
> > >   - **Line 50:** We identify the limitations of current methods (SFT and existing RL), noting that *"current training methodologies used to build these agents are simple and suboptimal."*
> > >   - **Line 53:** We further state, *"most of them focus solely on UI grounding... constrained by simplistic reward functions and naive algorithmic adaptations."* Consequently, we propose **"Progressive Reinforcement Learning"** as a more comprehensive strategy.
> > >
> > > - **On "Identified several seemingly simple yet important issues... proposed targeted solutions":**
> > >   Each module described in **Sec. 3.4** serves as evidence for this. For instance, the **Progressive Shrinking mechanism**, while seemingly simple, effectively improves the model's ability to perform fine-grained localization.
> > >
> > > ---
> > >
> > > > ### **Response to Q2: What DRIVES these two innovations?**
> > >
> > > In summary, these two innovations were driven by the continuous discovery of limitations during our experiments with existing GUI agents, leading us to design targeted solutions.
> > >
> > > **1. The Driver for Structured CoT:**
> > > - **Observation:** While using other open-source agents, we noticed their reasoning processes were often too brief and lacked explanation for their actions.
> > > - **Human Simulation:** To determine the ideal reasoning for GUI tasks, we manually performed these tasks and recorded our own thought processes (considering various factors before each click).
> > > - **Internalizing Multi-Agent Strengths:** We also observed that Multi-Agent Frameworks (which employ specialized experts for planning, reflection, perception, etc.) generally outperform end-to-end single agents.
> > > - **Solution:** We sought to incorporate this robust "expert workflow" into a single end-to-end model. By combining our human thought records with the multi-agent workflow concept, we developed **Structured CoT**. It forces the model to consider specific aspects, effectively internalizing the workflow into its reasoning process.
> > >
> > > **2. The Driver for Progressive RL:**
> > > Our training strategies were evolved to solve specific technical hurdles:
> > > - **Cold Start:** Improvements here were derived through multiple comparative experiments.
> > > - **Progressive Shrinking Mechanism:**
> > >   - *Initial Attempt:* We fixed the reward threshold at 14%, but the trained model struggled to interact precisely with small UI elements.
> > >   - *Second Attempt:* We lowered the threshold to 4%. However, valid rewards became too sparse in the early training stages, leading to instability and convergence difficulties.
> > >   - *Solution:* This tradeoff drove us to propose **Progressive Shrinking**, which we validated experimentally to solve both stability and precision issues.
> > > - **History Self-Alignment:** Empirically, we noticed a performance gap in long-horizon tasks caused by the inconsistency between using Ground Truth history (during training) and Model-Predicted history (during inference). **History Self-Alignment** was proposed specifically to bridge this gap.
> > > - **Corrective Teaching Pipeline:** After evaluating the trained model, we discovered some unforeseen errors (e.g., failing to hide the keyboard, not clearing text bars before typing, or failing to long-press for copy/paste). We developed this semi-automated pipeline to collect remedial data and specifically correct these behaviors.

---

> > > > ### Author Response · Authors · 2025-11-21
> > > > **Replying to Reviewer mgKA (2/2)**
> > > >
> > > > > ### **Response to Q3: Consensus on critical visual information**
> > > >
> > > > Thanks for pointing this out. We agree with your assessment. In the manuscript, the phrase *"extract critical information relevant to the task instruction"* is somewhat vague. In the revised version, we will provide a more precise and detailed description of what constitutes "critical visual information" to ensure readers clearly understand the methodology.
> > > >
> > > > ---
> > > >
> > > > > ### **Response to Q4: Point out the description of setup 'we leverage the strong general world knowledge of Gemini-2.5 Pro...' is stated in the manuscript?**
> > > >
> > > > This is stated in **Appendix C.1 (ANNOTATIONS METHOD)**, specifically on **Line 687**: *"To construct our training dataset, we employ Gemini-2.5-Pro, one of the most powerful VLM ......"*

---

> > > > > ### Comment · Reviewer_mgKA · 2025-11-21
> > > > > **Second Response to Following Author Response**
> > > > >
> > > > > Thank you for your response.
> > > > >
> > > > > On your Response to Q4:
> > > > > Line 687 states
> > > > > > To construct our training dataset, we employ Gemini-2.5-Pro (Comanici et al., 2025), one of the
> > > > > most powerful VLM, to generate step-by-step structured reasoning. Our annotation methodology is
> > > > > trajectory-level, designed to mirror the model’s sequential training and inference flow. Specifically,
> > > > > the reasoning generated for a given step is fed as historical context into the annotation process for the
> > > > > subsequent step.
> > > > >
> > > > > Do you consider it translates to your Initial response on Q1? Quoted below.
> > > > >
> > > > > > During data construction, we leverage the strong general world knowledge of Gemini-2.5 Pro, prompting it to generate necessary subtasks for each goal task. It is worth noting that the process is not simply splitting the goal task text into several subtask texts. Instead, the model is provided with the ground-truth trajectory (each step’s state screenshot + executed action). This ensures that each generated subtask is more accurate and reasonable.
> > > > >
> > > > > If you have other texts / materials describing this ('During data construction...') in your original manuscript. Can you point them out? If not, this is the missing explanation (in details) for my original question about the relationships between tasks and subtasks, which should be added to the paper.
> > > > >
> > > > > I don't have further questions at this moment.

---

> > > > > > ### Author Response · Authors · 2025-11-21
> > > > > > **Reply to the Reviewer mgKA**
> > > > > >
> > > > > > Thank you for your follow-up response.
> > > > > >
> > > > > > Regarding the original **Q1 ("How are subtasks derived from tasks? Is there a one-to-many relationship between tasks and subtasks? Are subtasks independent of each other?")**, we provided a detailed example to address each aspect. We hope this example successfully clarified the relationship and distinction between subtasks and goal tasks.
> > > > > >
> > > > > > However, we think that neither the original Q1 nor the supplementary questions explicitly requested a detailed description of how we utilized Gemini-2.5-Pro for data construction. Consequently, our responses focused on the definitions and relationships of subtasks, rather than the data pipeline itself.
> > > > > >
> > > > > > We acknowledge that the current Appendix C.1 in the original manuscript does not yet fully detail the data construction pipeline. We will ensure that the complete workflow and specific details of how data is generated are added to the revised version of the paper.
> > > > > >
> > > > > > *We hope our response has addressed your concerns. If you feel that the issues have been resolved, we would be grateful if you could consider raising your score. Thank you very much.*

---

### Official Review · Reviewer_VG1Y · 2025-10-31

**Soundness:** 1
**Presentation:** 2
**Contribution:** 1
**Rating:** 2
**Confidence:** 3

**Summary:**

This paper introduces MobileWizard, a data-efficient mobile GUI agent that combines structured reasoning and progressive reinforcement learning. The model decomposes reasoning into four interpretable modules, self-verification, screen analysis, planning, and action guidance, and employs progressive RL with reward shrinking, history self-alignment, and corrective teaching for continual improvement. Trained on only 24.5k public and 300 remedial trajectories, MobileWizard-7B achieves a strong performance on benchmarks, surpassing larger models.

**Strengths:**

1. The paper is well-structured and easy to follow.
2. MobileWizard has strong data efficiency. It achieves competitive or state-of-the-art results using only 24.5k public trajectories and 300 remedial ones.

**Weaknesses:**

1. The improvement on ScreenSpot-v2 is marginal. MobileWizard-7B achieves 93.0%, while its base model already reaches 92.4%, raising questions about the claimed gains.
2. Training dynamics are not illustrated; no learning curves or convergence analyses are provided, making it difficult to assess training stability.
3. Code and data are not provided, which limits reproducibility and transparency.
4. Using GPT-4o as a baseline is questionable; comparisons to stronger agent-based systems such as Operator or Claude Computer Use would be more appropriate.
5. The paper lacks comparisons with other RL-based GUI agents such as DigiRL and GUI-R1, which are directly relevant baselines.
6. The paper does not report performance after the cold-start phase, making it difficult to quantify the improvement brought by RL.

**Questions:**

1. The structured reasoning data are generated using Gemini-2.5 Pro, yet its performance is not reported in the benchmarks. Why is GPT-4o used instead?
2. Will the model, data, and training code be open-sourced?

---

> ### Author Response · Authors · 2025-11-21
> **Response to Reviewer VG1Y**
>
> ### **Response to W1: The improvement on ScreenSpot-v2 is marginal. MobileWizard-7B achieves 93.0%, while its base model already reaches 92.4%, raising questions about the claimed gains.**
> ScreenSpot-v2 is a relatively simple grounding benchmark, which cannot fully reflect a model's grounding capability. This feature is also mentioned in the UI-TARS paper. On the MMBench-GUI benchmark, MobileWizard-7B scores 79.8%, which is 31.8% higher than the baseline (48.0%). However, grounding ability is not the main focus of our work. We are primarily focus on the model's task success rate. Our proposed method shows a significant improvement over the baseline, ultimately achieving performance comparable to SOTA models of similar scale.
>
> ### **Response to W2: Training dynamics are not illustrated; no learning curves or convergence analyses are provided, making it difficult to assess training stability.**
> We have added relevant training dynamics analysis and corresponding figures in Appendix D of the newly updated paper. As shown, the loss steadily declines and then plateaus in both the cold-start and RL stages. During the RL stage, it can be observed that the reward on the validation set increases gradually. This demonstrates that the training stability of MobileWizard is well maintained throughout the process.
>
> ### **Response to W3：Code and data are not provided, which limits reproducibility and transparency.**
> The code and data will be released in the open-source and camera-ready versions.
>
> ### **Response to W4：Using GPT-4o as a baseline is questionable; comparisons to stronger agent-based systems such as Operator or Claude Computer Use would be more appropriate.**
> OpenAI's Operator does not currently support interaction on mobile systems, so a direct comparison is not possible. We have provided a comparison with the relevant Computer Use models as follows:
> | | AndroidWorld (SR %) |
> | :--- | :--- |
> | OpenAI CUA o3 | 52.5 |
> | Claude Computer Use (Sonnet 4.5) | 56.0 |
> | MobileWizard–7B | 47.2 |
>
> As results shown, the performance gap between MobileWizard-7B and such closed-source SOTA expert models is not substantial. We will subsequently focus on surpassing them by scaling up both data, model, and training.
>
> ### **Response to W5：The paper lacks comparisons with other RL-based GUI agents such as DigiRL and GUI-R1, which are directly relevant baselines.**
> For GUI-R1, our Table 3 already contains a comparison with it.
>
> For DigiRL，we provide more experiments as follows:
> | | AitW General – Test | AitW Web Shopping – Test |
> | :--- | :--- | :--- |
> | DigiIRL | 71.9 | 67.2 |
> | MobileWizard | 73.3 | 69.5 |
>
> ### **Response to W6：The paper does not report performance after the cold-start phase, making it difficult to quantify the improvement brought by RL.**
> Thanks for pointing this out. Here we provide additional experimental results, including those from cold-start experiments with varying amounts of data.
> | AndroidWorld (SR) | 5% | 25% | 50% | 75% | 100% |
> | :--- | :--- | :--- | :--- | :--- | :--- |
> | cold start | 30.0 | 32.1 | 32.9 | 33.4 | 34.5 |
> | cold start + 1epoch RL | 33.3 | 37.5 | 35.2 | 35.6 | 36.7 |
>
> ### **Response to Q1：The structured reasoning data are generated using Gemini-2.5 Pro, yet its performance is not reported in the benchmarks. Why is GPT-4o used instead?**
> GPT-4o is a closed-source baseline model that is used for comparison in almost all GUI-Agent works. Its results have been widely reproduced and validated by researchers, making them highly credible and convincing. Here, we additionally provide our own results obtained by running Gemini 2.5 on AndroidWorld:
> | | AndroidWorld (SR %) |
> | :--- | :--- |
> | gemini-2.5–flash - m3a | 43.2 |
> | gemini-2.5–pro - m3a | 50.7 |
> | MobileWizard–7B | 47.2 |
>
> ### **Response to Q2：Will the model, data, and training code be open-sourced?**
> Yes, all relevant resources—including the model, data, and training/inference code—will be made publicly available in the open-source and camera-ready versions.

---

> > ### Comment · Reviewer_VG1Y · 2025-11-27
> >
> > Thanks for the clarification, which addresses part of my concerns. I have updated my score accordingly, but several issues remain:
> >
> > 1. The proposed workflow consists of several modules and reward design, yet the performance improvement is rather limited. Notably, the model performs best when trained with only 25% of the SFT data, which raises my great concerns about the scalability of the method. If the method were truly general and robust, additional data should lead to further performance gains.
> >
> > 2. Please avoid directly including TensorBoard screenshots in the paper, which is quite unprofessional.
> >
> > 3. It is evident that both the rebuttal and the paper were generated or at least heavily polished using LLMs. Please at least revise the text to remove the patterns introduced by such tools.

---

> > > ### Author Response · Authors · 2025-11-28
> > > **Reply to Reviewer VG1Y**
> > >
> > > Thanks for your comments.
> > >
> > > 1. Regarding Scalability: Achieving the best results by using 25% of the data for the cold-start phase before RL does not imply poor scalability. Instead, it highlights an optimization strategy: how to best allocate data between the cold-start and RL phases. Our findings suggest that the cold-start phase doesn't require massive data—it just needs part of them to establish basic reasoning capabilities. The remaining data can then be used for RL to maximize performance. While the optimal ratio might shift as the dataset size grows, the methodology itself remains scalable.
> > >
> > > 2. Regarding TensorBoard: TensorBoard is a standard, widely-accepted tool for visualizing training dynamics. We respectfully disagree that using it is "unprofessional," particularly during the short rebuttal time. While we could have redrawn every plot for better aesthetics, the underlying results would remain exactly the same. Given the time constraints, we prioritized presenting the core data and results over creating polished graphics. We hope you understand.
> > >
> > > 3. Regarding LLM Usage: We acknowledge using LLMs to check grammar and polish our phrasing. However, we strongly refute the claim that the paper or this rebuttal were "generated" or "heavily polished" by LLMs. If you maintain this view, we ask that you identify all content that seem artificial to you and suggest how they should be rewritten. This would help us understand exactly what standard of expression you are looking for.

---

> ### Comment · Area_Chair_MMNY · 2025-11-25
> **Please participate in discussions with authors and other reviewers asap**
>
> Please ensure you are actively participating in the discussion phase.
>
> Additionally, I strongly encourage you to read the other reviews and discuss with your fellow reviewers. It is vital to compare perspectives and raise any remaining concerns now to give the authors a fair opportunity to respond.
>
> Based on these interactions, please update your reviews and finalize your decisions.
>
> Best, AC

---

### Official Review · Reviewer_xksQ · 2025-11-04

**Soundness:** 2
**Presentation:** 3
**Contribution:** 2
**Rating:** 6
**Confidence:** 5

**Summary:**

The paper presents and agent framework designed for mobile GUI manipulation using data more efficiently and also leveraging remedial trajectories to improve agent adaptation to failures. The proposed framework pitches a small model/small data approach as scalabe and effective for training mobile agents.

The framework is composed of two core pieces:
1) a CoT-like structured reasoning apporach, including four sub-modules: screen analysis, planning, action guidance, and self-verification outputing a trajectory containing the four units per step.
2) a Progressive Reinforcement Learning (PRL) procedure, itself also divided into four components: cold-start training, dynamic reward system, history self-alignment, and a Corrective Teaching Pipeline.

PRL emphasizes the effective use of data for cold start, a four-parts reward function, self-alignment for efficient use of part history during inference, and a corrective teaching pipeline - were external LMMMs are used to classify failures and propose correct actions, which are then used as remedial trajectories back in the RL loop. The paper claims each components contribute for the effective usage of as little data as possible.

Results on two UI grounding benchmarks show the framework can reach SOTA results or competitive results for models of the same size (7B) while using much less data. Further task execution benchmark results in two static benchmark and a dynamic benchmark further illustrate framework performance, showcasing its competitive performance.

The author also provide ablations, using the AndroidWorld benchmark, for four key components in the framework, which show their impact in improving perfomance versus simpler versions of the component or disabling each components in the framework.

**Strengths:**

The proposed framework is pragmatically designed and achieves strong UI grounding results in competitive agent benchmarks, as well as competitive performance in the dynamic task execution AndroidWorld benchmark, even using less data than other similar sized models.

The paper presents and interesting system integration of different components that applies RL techniques in task execution scenarios while trying to reduce/minimize data use for more effective agent training.

Experimental results suggest this strategy may have promise.

**Weaknesses:**

While a complex system per se, or relative small gains individually in benchmark results, are not an issue if the overall system is effective, however the proposed framework includes two many new pieces at once (2 core modules, divided each into 4 sub-modules, with a 4 component reward function, and two-step corrective pipeline using external models) that make it hard to properly analyse and determine if the presented result support its key claims. A lot of the paper space is used describing the system, but little on the analysis of each component or on the claims or effective use of little data.

Regarding the structured CoT-like, which is only evaluated as a single block in ablation, differ from other agent work that also perform "self-verification, screen analysis, planning, and action guidance"? They are not novel concepts per se, and, for example, [1] presents an agent that does "self-reflection, information gathering, task inference, and action planning", which seem pretty analogous to the proposed framework. [1] also represents this output as structured memory/history fed to the model. The novelty claims and benefits here don't seem well supported (it increases MobileWizard performance less then 2 percent points in SR).

Section 3.4 seems to be the core contribution of the paper, but also here the contributions of each module are not well supported by the experiments. Regarding performance improvement in the presented ablation, each individually contribute only around 1.5 to 2 percent point to overall results.

More importantly, for the core claim of the paper, reduced data usage, no experiments are provided for different settings and few details are giving for the current used approach. For example, when discussing the optimization of the initial cold-start training data volume, the paper claims 25% of the data was best. Based on how diverse runs? Only one uniform sample? If the same evaluation was used for all sample rounds, how was evaluation coverage/generality guaranteed?

The presented reward function is also complex, with 4 components, some somewhat trivial and some seemingly not well justified in their parameters. For example, in Action Type reward from groundtruth, how does this encourage different solutions if events must match the specific action in the trace? It seems somehow spurious, especially as parameters are its own reward component next. Also for the progressive shrinking mechanism. Gradually tightening the threshold seems very interesting, but why the chosen range though? Finally it reaches 4%, but does one know it's good enough? How does it affect UI's that require precision? How well are such cases represented in the benchmark data?

The 4th reasoning reward requires very detailed annotated training data for format of reasoning and uses only Sentence-BERT as metric. This can also be significant if really having little high-quality data is key. But there is no in-depth analysis of how much data is actually need and, more importantly, how it scales. Similar comments apply to history self-alignment and how it's difference fromthe UI-Venus approach not negatively affect peformance. Or also to the corrective pipeline. How do the chosen traces help generalization? 300 are really sufficient?

In summary, the paper is interesting, I like the system side of it and it achieves good results. But the design decisions are not well backed in the manuscript, and it doesn't provide enough evidence for its core claim of a "scalable and efficient path toward building more robust and versatile GUI agents". Especially in scaling and generalization ability. If toning down these claims it is still good work.

[1] Weihao Tan et al. CRADLE: Empowering Foundation Agents Towards General Computer Control. ICML 2025.

**Questions:**

In the corrective teaching pipeline, the paper claim to really run apps under a simulator for corrective teaching. How do you make sure of the correct app state?

GUI-Odissey and AC are part of the training data, so that can partially account for the performance in the offline benchmark cases in Table 3. How does this data correlate to AndroidWorld? The performance there does look interesting.

Why would you say a larger UI-TARS leads to worse results in ScreenSpot and MMBench-ui? Does the same hold for UI-TARS 2?

There are also some presentation issues to fix:
- In Table 2, performance in grounding, there is incorrect use of underline to mark second best. Some columns don't have best/second, and some the underlined is actually the best.
- In Table 3, SEED-VL and UI-TARS are not from OpenAI and at least UI-TARS is not a closed model. Also, if similar performance is going to be considered equal for best/second, define the deviation and be consistent in the whole table.
- Xu et al. 2024 is not the best example of early image only agent input. Other work predates it on arxiv and published.

---

> ### Author Response · Authors · 2025-11-21
> **Response to Reviewer xksQ (1/3)**
>
> ### **Response to W1: includes two many new pieces at once in the framework that make it hard to properly analyze and determine if the presented result support its key claims.**
> Thanks for your comments. From a high-level perspective, MobileWizard can be divided into two main components: how to build structured reasoning capabilities, and how to conduct efficient RL training. Although each part consists of multiple modules, they can be viewed systematically. As presented in the paper, these components are coupled together and progressively enhance MobileWizard’s performance. Ablation studies in Section 4.3 illustrate the performance gains brought by structured reasoning and progressive RL training compared to the baseline. In other words, using the same dataset, the integration of these two components progressively increased the success rate on the AndroidWorld benchmark to 47.2%, offering a highly effective reasoning and training framework for future research.
>
> ### **Response to W2: The novelty and benefits about the structured CoT**
> Thank you for pointing this out. You are correct that workflows such as self-verification, screen analysis, planning, and action guidance have frequently been implemented in agent frameworks, where each specialist handles one part, forming a collaborative system.
>
> However, the key difference with MobileWizard is that it is a native single end-to-end model designed to possess similar workflow capabilities. A central idea is to embed this kind of workflow within the model's reasoning process. Regarding memory/history construction: unlike an agent framework that can directly integrate outputs from different specialists to form memory, an end-to-end model must build memory from its own outputs. When the reasoning process explicitly includes content for each step, constructing memory naturally becomes straightforward.
>
> As for the benefits, in ablation study (a) where models were trained for only 1 RL epoch, structured reasoning already demonstrated a 2.1% performance improvement. Due to constraints in computational resources and time, we were unable to train a free-form CoT model from scratch for a full comparison. However, we believe that an initial model with better reasoning capabilities can achieve greater gains through longer-term progressive RL training.
>
> # **Response to W3: each individually contribute only around 1.5 to 2 percent point to overall results**
> In our ablation studies, due to resource constraints, we employed a controlled variable approach and trained each contribution for only one epoch. This allowed us to independently and quickly validate the effectiveness of each module. The performance improvement over the baseline is expected to become more pronounced with additional RL training epochs. Here we provide additional experiments for clarification:
> | | AndroidWorld (SR %) – 1 turn | AndroidWorld (SR %) – 5 turn |
> | :--- | :--- | :--- |
> | w/o HSA | 38.0 | 41.3 |
> | w/ HSA | 40.8 | 45.6 |
>
> ### **Response to W4: For the core claim of the paper, reduced data usage, no experiments are provided for different settings are giving for the current used approach. The paper claims 25% of the data was best. Based on how diverse runs? Only one uniform sample? If the same evaluation was used for all sample rounds, how was evaluation coverage/generality guaranteed?**
>
> Thank you for pointing this out. We would first like to clarify an important aspect: our primary goal is to explore whether there are effective ways to improve model performance through model reasoning and training strategies, beyond simply increasing the volume of training data. However, we are not advocating for reduced data usage. If we could easily collect more high-quality training data, it would undoubtedly benefit the model's performance. Therefore, data remains an important factor. What we propose are methods that achieves better performance using the same amount of data.
>
> Regarding the cold-start training, we conducted three rounds of experiments, each time sampling different subsets of data while maintaining a uniform sampling strategy. For evaluation, since our ultimate goal is to improve the task success rate (SR), we used online benchmarking platforms such as AndroidWorld. We also tested on the offline benchmark AndroidControl—results are shown below. It can be observed that under MobileWizard’s training data and settings, using around 25% of the data for cold-start before 1 epoch RL yields better initial performance.
> | | 5% | 25% | 50% | 75% | 100% |
> | :--- | :--- | :--- | :--- | :--- | :--- |
> | AndroidControl–High (SR) | 71.9 | 73.1 | 72.5 | 72.7 | 73.0 |

---

> > ### Author Response · Authors · 2025-11-21
> > **Response to Reviewer xksQ (2/3)**
> >
> > ### **Response to W5: For Action Type reward: How does it encourage diverse solutions if events must match trace-specific actions? And, progressive threshold tightening—why this range? Does reaching 4% suffice? How does it impact precision-requiring UIs?**
> >
> > Regarding the Action Type reward, due to limitations in the dataset (which lacks annotations for multiple valid action paths), the rule-based reward currently relies on the single available correct action annotation. This reflects a broader challenge in the GUI-Agent field, where effective solutions for multi-path scenarios are still lacking. However, our experimental results show that even without multi-path annotations to provide additional reward signals, the model can still learn effectively.
> >
> > For the progressive shrinking schedule (14% -> 10% -> 7% -> 4%), our experiments show that for the first three thresholds, a variation of ±1% has minimal impact. However, the final threshold is more critical. If set too low, the model would only receive a reward for clicking the exact pixel, which contradicts real-world interaction patterns. This would make rewards extremely sparse, leading to difficult training conditions and preventing further performance improvements.
> > | | AndroidWorld (SR %) |
> > | :--- | :--- |
> > | 4% | 45.6 |
> > | 2% | 44.8 |
> > | 1% | 42.9 |
> >
> > ### **Response to W6: The 4th reasoning reward needs highly detailed annotation—no analysis of required data quantity/scalability. Same for history self-alignment: how do differences from UI-Venus not hurt performance? And corrective pipeline: do 300 traces ensure generalization?**
> >
> > - Regarding "how much data is actually needed": if the goal is to achieve over 70% success rate, the public dataset used in this paper is far from sufficient. However, collecting high-quality GUI agent trajectory data is often costly. Thus, this work starts with a limited dataset and explores how to improve performance through reasoning structures and training strategies, laying the groundwork for future scaling.
> >
> > - As for "how it scales": with a robust infrastructure where human annotators and machines collaborate semi-automatically—collecting data from both real devices and simulators, and using a powerful VLM to auto-annotate reasoning content (as described in Appendix C.1)—we can scale the production of detailed annotated training data. This is a direction we plan to pursue.
> >
> > - For history self-alignment, Sec. 3.4.3 details the differences from UI-Venus, mainly our dynamic reward-based triggering mechanism and trajectory-level alignment. While both methods use the model’s own reasoning output as history during training, our approach is more efficient. Below is a comparison between UI-Venus and MobileWizard:
> > | | AndroidWorld (SR %) |
> > | :--- | :--- |
> > | HSA from UI–Venus | 39.9 |
> > | HSA from MobileWizard | 40.8 |
> >
> > - Regarding the corrective pipeline, the chosen traces are constructed from the model’s own failure trajectories. These traces are highly targeted and instructive—acting like a teacher showing how to handle similar situations—thus aiding generalization.
> > As for the remedial data, the 300 traces are limited due to resource, time, and annotation constraints, and are indeed insufficient. However, the fact that we observe preliminary improvements with only 300 traces suggests this method has promising potential for scaling, which we will explore in future work.

---

> > > ### Author Response · Authors · 2025-11-21
> > > **Response to Reviewer xksQ (3/3)**
> > >
> > > ### **Response to Q1: In the corrective teaching pipeline, the paper claim to really run apps under a simulator for corrective teaching. How do you make sure of the correct app state?**
> > >
> > > The Corrective Teaching Pipeline is a semi-automated process developed through collaboration between humans and models. Specifically:
> > >
> > > 1. We designed a specialized prompt template for trajectory judgment, prompting Gemini-2.5-Pro to evaluate whether an entire trajectory is successful based on multiple criteria. Successful trajectories are filtered out, while only failed trajectories are retained.
> > > 2. For failed trajectories, we identify the specific steps that caused the failure (manually or using a VLM). Then, UI-TARS-1.5 takes over from MobileWizard to complete the remaining steps.
> > > 3. The newly completed trajectory is judged again by Gemini-2.5-Pro. Only trajectories deemed successful at this stage proceed to the next step.
> > > 4. Based on the data collected in Step 3 and the confidence scores assigned by the VLM, human annotators perform a secondary review to ensure the collected trajectories are correct and effective.
> > >
> > > ### **Response to Q2: In the corrective teaching pipeline, the paper claim to really run apps under a simulator for corrective teaching. How do you make sure of the correct app state?**
> > >
> > > GUI-Odyssey and AC are both high-quality datasets that positively contribute to performance on the AndroidWorld benchmark. Constrained by limited resources, we conducted a quick experiment comparing three settings: using 100% of the SFT data, removing GUI-Odyssey from SFT, and removing AC from SFT. The results show that removing either dataset leads to a drop in accuracy.
> > > | | AndroidWorld (SR %) |
> > > | :--- | :--- |
> > > | SFT | 35.6 |
> > > | SFT w/o GUI–O | 33.9 |
> > > | SFT w/o AC | 33.5 |
> > >
> > > ### **Response to Q3: Why would you say a larger UI-TARS leads to worse results in ScreenSpot and MMBench-ui? Does the same hold for UI-TARS 2?**
> > > This is the result reported in the UI-TARS paper. The authors indicate that ScreenSpot v2 may not fully reflect the advantages of larger models in grounding capability, as UI-TARS-72B significantly outperforms UI-TARS-7B on the more complex and demanding ScreenSpot Pro benchmark. This suggests that the benefits of the 72B model's scale become more evident when handling highly complex, information-dense GUI scenarios. For UI-TARS 2, since models of different sizes are not publicly available, it remains unclear whether the same phenomenon exists.
> > >
> > > ### **Response to Q4: Some presentation issues to fix**
> > > We sincerely appreciate you raising these points. We will address all the presentation issues in our revised version.

---

> ### Comment · Area_Chair_MMNY · 2025-11-25
> **Please participate in discussions with authors and other reviewers asap**
>
> Please ensure you are actively participating in the discussion phase.
>
> Additionally, I strongly encourage you to read the other reviews and discuss with your fellow reviewers. It is vital to compare perspectives and raise any remaining concerns now to give the authors a fair opportunity to respond.
>
> Based on these interactions, please update your reviews and finalize your decisions.
>
> Best, AC

---

### Official Review · Reviewer_EwP9 · 2025-11-04

**Soundness:** 3
**Presentation:** 3
**Contribution:** 3
**Rating:** 6
**Confidence:** 4

**Summary:**

MobileWizard is a 7B mobile GUI agent trained with (i) a structured reasoning format and (ii) a progressive RL recipe, comprising of a multi-part reward with progressive thresholds. The model was trained on 24.5k public trajectories and 300 remedial ones. MobileWizard reports 47.2% success-rate on AndroidWorld outperforming a 72B UI-TARS model. Further, it ties 93.0 on ScreenSpot-v2 with UI-TARS 7B.

**Strengths:**

- Paper is clean and easy to understand.
- Progressive shrinking of the reward threshold is a sensible way to avoid reward hacking in later stages of the training while keeping the task easy enough for the model to learn.
- Strong performance on benchmarks compared to existing state-of-the-art models.

**Weaknesses:**

- Missing information on test-time compute for the baselines against MobileWizard. Inference-time reasoning length, steps per episode, and latency are not reported or compared to baselines (e.g., UI-TARS, OS-Atlas, UI-Venus).

- Although progressive shrinking seems reasonable, I am curious to know how would the model perform with a fixed threshold.

- Nit: Increase the font size in figures. (e.g., Figure 3)

**Questions:**

- Does the model need a RL agent to learn? Curious to know, how would the training curve and performance looks like if the model was only trained on high-quality SFT data.

- What are the failure modes that arise from a baseline RL setup that MobileWizard solves? Similarly, what are the failure modes that still exist in MobileWizard.

---

> ### Author Response · Authors · 2025-11-21
> **Response to Reviewer EwP9**
>
> ### **Response to W1: Missing information on test-time compute for the baselines against MobileWizard. Inference-time reasoning length, steps per episode, and latency.**
> Thanks for pointing this out. We have added details on the compared methods' input history format, reasoning length (CoT), and inference latency for both the initial and later stages. MobileWizard's longer CoT leads to slightly slower initial step latency, while UI-TARS is the slowest in later steps due to processing the most history, including past images. In contrast, UI-Venus and MobileWizard exhibit similar, faster inference speeds in later steps.
> | | History Format | CoT Length each step | latency for init step | latency for last steps |
> | :--- | :--- | :--- | :--- | :--- |
> | UI-TARS-1.5–7B | text + past images | 70–140 | ~1000ms | ~3700ms |
> | UI–Venus–Navi–7B | text | 80–150 | ~1000ms | ~3000ms |
> | MobileWizard–7B | text | 110–200 | ~1200ms | ~3000ms |
>
> ### **Response to W2: how would the model perform with a fixed threshold.**
> Thanks for raising this important issue. During experiments, we found that using a fixed threshold (e.g., 14%) caused localization failures on small UI elements during online evaluation, while a strict threshold (e.g., 4%) led to sparse rewards and prevented the initial RL training from converging effectively.
> | Method | AndroidWorld (SR %) |
> | :--- | :--- |
> | Fix (4%) | 44.2 |
> | Fix (14%) | 42.9 |
> | progressive shrinking | 47.2 |
>
> ### **Response to W3: (Nit:) Increase the font size in figures.**
> Thanks for your feedback on the details presented in the article. We will address these issues thoroughly in the revised version.
>
> ### **Response to Q1: Does the model need a RL agent to learn? How would the training curve and performance looks like if the model was only trained on high-quality SFT data.**
> Yes. The experimental results indicate that RL training is an effective strategy, as it strengthens the accuracy of the Agent's reasoning path in a single sampling. We also conducted an experiment using only high-quality SFT data, with the following results:
> | Method | AndroidWorld (SR %) |
> | :--- | :--- |
> | Only SFT with 1 epoch | 35.6 |
> | Only SFT with 2 epochs | 36.8 |
> | Only Cold_Start with 25% data | 32.1 |
> | Cold_Start + Progressive RL | 47.2 |
>
> We provide the training curves in Appendix D of the newly updated paper. These results demonstrate that SFT alone is insufficient for continuously enhancing the Agent's reasoning and decision-making capabilities. In contrast, applying our progressive RL training after a cold start can stimulate the Agent's reasoning potential, leading to improved decision-making performance.
>
> ### **Response to Q2: What are the failure modes that MobileWizard solves? What are the failure modes that still exist in MobileWizard.**
> - Failure Modes from a Baseline RL Setup and How MobileWizard Addresses Them:
>
> (1) Long-tail action distribution hindered access to effective rewards for rare actions (like Answer or Long Press); this is resolved by our proposed efficient data distribution (Sec. 3.4.1).
>
> (2) Difficulty interacting with small UI elements (as noted in Weakness 2) is effectively solved by the Progressive Shrinking method (Sec. 3.4.2).
>
> (3) A train/inference gap arose because the Agent used ground-truth history during RL training but self-generated history during online evaluation; our history self-alignment technique closes this gap, boosting AW benchmark performance by 2.8% (Sec. 3.4.3).
>
> (4) Unforeseen simple errors during online execution: In real-world tasks, simple but unpredictable errors could occur—like failing to zoom after waking the keyboard, not clearing an input field, or missing a line break.
> Solution (Sec. 3.4.4): The Corrective Teaching Pipeline teaches the Agent how to handle such situations, reducing the probability of getting stuck due to invalid operations.
>
> - Failure modes that still exist:
>
> (1) Difficulty in extracting important visual information from long text passages: MobileWizard often fails at tasks that require extracting specific information from long passages of text and then memorizing it to fill in relevant fields. The primary issue is its inability to accurately identify and retain the correct information from the text, leading to errors in subsequent steps.
>
> (2) Limited generalization to apps with novel layouts and functionalities: When performing tasks in new apps whose layout and functionality differ significantly from the samples seen during its training, MobileWizard frequently makes errors. This highlights a limitation in its ability to generalize.
>
> (3) In the training dataset, there are very few examples of horizontal scrolling but an abundance of vertical scrolling samples, a result of the specific scenarios and tasks used for data collection. Consequently, MobileWizard has largely failed to learn when and how to perform horizontal scrolls.

---

> ### Comment · Area_Chair_MMNY · 2025-11-25
> **Please participate in discussions with authors and other reviewers asap**
>
> Please ensure you are actively participating in the discussion phase.
>
> Additionally, I strongly encourage you to read the other reviews and discuss with your fellow reviewers. It is vital to compare perspectives and raise any remaining concerns now to give the authors a fair opportunity to respond.
>
> Based on these interactions, please update your reviews and finalize your decisions.
>
> Best, AC

---

> > ### Comment · Reviewer_EwP9 · 2025-11-27
> >
> > Thank you authors for the rebuttal.
> > I have a follow-up question. For W1 (Missing information on test-time compute for the baselines against MobileWizard), I am curious to know if the baseline models had equal token-budget or episode-budget to compare against MobileWizard. How does test-time compute normalized numbers look?

---

> > > ### Author Response · Authors · 2025-11-28
> > > **Reply to Reviewer EwP9**
> > >
> > > Thanks for your reply.
> > >
> > > 1. **Token Budget**: We set the budget to 32768 for all benchmarks (online and offline). We applied this to both MobileWizard and the reproduction baseline methods . This token budget is high enough to handle all agent's thinking and output without cutting anything off, ensuring a fair comparison.
> > >
> > > 2. **Episode Budget**: For AndroidWorld, every task has a strict max-steps limit depending on how hard it is. All agents tested on it using these same limits to keep the episode budget consistent.

---

### Meta-Review · Area_Chair_RDZF · 2026-01-01

**Summary:**

This paper proposes MobileWizard, a mobile GUI agent trained with 24.5k public trajectories and 300 remedial trajectories. The method combines (1) Structured Reasoning, a structured Chain-of-Thought with four modules (screen analysis, planning, action guidance, self-verification), and (2) Progressive Reinforcement Learning (PRL), comprising cold-start training, progressive reward shrinking, history self-alignment, and a corrective teaching pipeline. The paper reports 47.2% success rate on AndroidWorld and claims improved performance over several open-source baselines, including larger models such as UI-TARS-72B. Reviewer opinions are mixed (two borderline-positive scores and two rejects). However, reviewers raise concerns about novelty/positioning, the level of evidence and analysis relative to the complexity of the proposed system, support for strong claims around data-efficiency/scalability/generalization, baseline framing/completeness, and reproducibility/presentation clarity. The rebuttal addresses several concrete issues (test-time compute details, fixed-threshold vs progressive shrinking ablations, added training curves, cold-start vs RL breakdown, and additional baseline results), but multiple reviewers’ core concerns remain. The authors also clarify in rebuttal (in response to xksQ) that they are not advocating reduced data usage, but rather proposing strategies to improve performance under a fixed data budget. Based on the full discussion, my recommendation is Reject.

**Reviewer Concerns:**

### Concerns substantially or partially addressed by the rebuttal

- Test-time compute and fairness of comparisons (EwP9): The authors added details on baseline history formats, CoT lengths, and inference latency, and clarified that token budgets (32768) and AndroidWorld step limits were consistent across methods.

- Necessity of progressive reward shrinking (EwP9, xksQ): The rebuttal includes fixed-threshold comparisons (4%, 14%) and shows progressive shrinking performs best, along with reasoning about reward sparsity and localization failures.

- Lack of training dynamics / stability evidence (VG1Y): The authors state they added training dynamics and corresponding figures (Appendix D) showing loss and reward progression.

- Cold-start vs RL contribution and additional baseline comparisons (VG1Y): The rebuttal adds cold-start vs RL breakdown results and provides additional experiments such as DigiRL comparisons.

### Outstanding concerns that remain insufficiently resolved

- Novelty and contribution positioning (xksQ, VG1Y, mgKA): Reviewers argue that the proposed structured reasoning modules closely resemble prior agent workflows, and the novelty claims are not convincingly supported through comparison and analysis in the manuscript. The authors argue a key difference is internalizing a multi-agent workflow into a single end-to-end model via structured reasoning outputs, but reviewers still find the novelty/positioning insufficiently established in the manuscript.

- System complexity vs. evidence and analysis (xksQ, VG1Y):  Reviewers note the method introduces many components (multiple PRL modules, multi-part reward design, corrective teaching pipeline using external models), while the per-module gains in ablations appear modest and the manuscript lacks sufficient analysis and justification of key design choices.

- Scalability / data-efficiency claims (xksQ, VG1Y, mgKA): Reviewers question whether the paper provides adequate evidence supporting strong claims about scalability and robust generalization, especially given limited scaling experiments and concerns raised about the “25% cold-start data performs best” observation. The rebuttal clarifies that the authors are not advocating reduced data usage; however, reviewers’ concerns remain regarding the level of evidence supporting stronger scalability/generalization claims.

- Baseline framing and comparison completeness (VG1Y, mgKA): Reviewer VG1Y questioned missing comparisons to DigiRL and GUI-R1; the rebuttal notes that GUI-R1 is already included in Table 3 and adds additional DigiRL results. Reviewer mgKA also questions strong wording given that some baselines outperform MobileWizard on specific benchmarks (e.g., UI-Venus-Navi-7B achieving 49.1% SR vs. 47.2%).

- Reproducibility (VG1Y):  Code and data are not currently released; the authors state they will release them in the open-source and camera-ready versions, but reproducibility remains limited at this stage.

- Presentation and writing issues (xksQ, mgKA, VG1Y):  Reviewers cite missing explanations for figures, table formatting and labeling errors, vague terminology (e.g., “critical visual information”), and concerns about presentation quality (e.g., TensorBoard screenshots).

**Reviewer Scores:**

Reviewer EwP9 (original score: 6): No explicit statement about changing the score in the discussion thread.

Reviewer xksQ (original score: 6): No explicit statement about changing the score in the discussion thread.

Reviewer VG1Y (original score: 2): Explicitly stated: “I have updated my score accordingly”, but did not specify the new score value in the discussion thread. Also, several concerns of this reviewer remain.

Reviewer mgKA (original score: 2): No explicit statement about changing the score in the discussion thread.

---

### Decision · Program_Chairs · 2026-01-26

Reject